# Automatic Meter Reading from UAV Inspection Photos in the Substation by Combining YOLOv5s and DeeplabV3+

**DOI:** 10.3390/s22187090

**Published:** 2022-09-19

**Authors:** Guanghong Deng, Tongbin Huang, Baihao Lin, Hongkai Liu, Rui Yang, Wenlong Jing

**Affiliations:** 1Guangzhou iMapCloud Intelligent Technology Co., Ltd., Guangzhou 510095, China; 2Guangdong Province Engineering Laboratory for Geographic Spatiotemporal Big Data, Key Laboratory of Guangdong for Utilization of Remote Sensing and Geographical Information System, Guangdong Open Laboratory of Geospatial Information Technology and Application, Guangzhou Institute of Geography, Guangdong Academy of Sciences, Guangzhou 510070, China

**Keywords:** object detection, image segmentation, YOLOv5s, Deeplabv3+, meter reading

## Abstract

The combination of unmanned aerial vehicles (UAVs) and artificial intelligence is significant and is a key topic in recent substation inspection applications; and meter reading is one of the challenging tasks. This paper proposes a method based on the combination of YOLOv5s object detection and Deeplabv3+ image segmentation to obtain meter readings through the post-processing of segmented images. Firstly, YOLOv5s was introduced to detect the meter dial area and the meter was classified. Following this, the detected and classified images were passed to the image segmentation algorithm. The backbone network of the Deeplabv3+ algorithm was improved by using the MobileNetv2 network, and the model size was reduced on the premise that the effective extraction of tick marks and pointers was ensured. To account for the inaccurate reading of the meter, the divided pointer and scale area were corroded first, and then the concentric circle sampling method was used to flatten the circular dial area into a rectangular area. Several analog meter readings were calculated by flattening the area scale distance. The experimental results show that the mean average precision of 50 (mAP50) of the YOLOv5s model with this method in this data set reached 99.58%, that the single detection speed reached 22.2 ms, and that the mean intersection over union (mIoU) of the image segmentation model reached 78.92%, 76.15%, 79.12%, 81.17%, and 75.73%, respectively. The single segmentation speed reached 35.1 ms. At the same time, the effects of various commonly used detection and segmentation algorithms on the recognition of meter readings were compared. The results show that the method in this paper significantly improved the accuracy and practicability of substation meter reading detection in complex situations.

## 1. Introduction

Meter reading is an extremely important task that is widely used in real life [1]. Many meter readings [2,3,4] require manual periodic inspection and recording of the meter readings to reflect whether the device is operating safely. Due to the various shapes, scales, pointers, and characters of the meter, and the existence of different equipment locations, such as high-voltage equipment in substations, it is difficult for regular manual inspections to be carried out.

At present, traditional meters with scales and pointers are still commonly used in the substation environment. In the actual application environment, the traditional manual counting not only involves many human factors, but also is easily disturbed by the environment and has the danger of an electric shock. With the gradual promotion of unattended substations, inspection robots or UAVs equipped with automatic meter identification technology have been widely used. Therefore, the realization of intelligent reading of meters by the computer vision method has become a research hotspot.

In recent years, the traditional meter reading methods have been mainly based on Hough transform [5,6,7,8] and image registration [9,10]. Firstly, the method based on Hough transform is to obtain the position of the pointer and the dial through Hough line detection and arc detection, and also to obtain the deflection angle of the pointer and calculate the reading of the meter. This method is susceptible to noise interference, resulting in inaccurate readings. Secondly, some researchers also use the method based on image template matching to identify the instrument. This method uses feature matching algorithms such as SIFT [11] and SURF [12] to register the image to be recognized as a standard image to read meter readings, and it is not conducive to dealing with multiple meter readings in complex backgrounds.

With the continuous development of deep learning and convolutional neural networks, many common object detection algorithms and semantic segmentation algorithms [13] are used in meter reading detection, such as YOLO [14,15], SSD [16], Faster RCNN [17], DeepLab [18], Unet [19], and Transformer [20], etc. In recent years, many researchers have used deep learning methods to conduct research in this field. Xing et al. [21] used a convolutional neural network to detect the area where the meter exists and post-process the meter image through a Hough transform. However, the robustness of the algorithm is not strong. Wan et al. [22] proposed a method based on a combination of object detection and image segmentation algorithms and a perspective transformation method based on image segmentation information to calibrate the image and finally obtain the meter reading. However, this method requires a two-stage detection of the dial and hands, which increases the calculation time. Tao et al. [23] used the SSD algorithm combined with the post-processing method to achieve meter readings, but this method has a large error, and the model size is also large and takes up a lot of memory.

Most of the above methods are not suitable for carrying on mobile terminals or UAVs, and there is the problem of large errors. In order to solve the above problems, this paper proposes a meter detection and recognition method based on UAV aerial images. Specifically, it is to deploy UAV slots in substations, plan routes, and collect meter images in different weather and time periods in substations. The meter area in the image is detected by using the YOLOv5 object detection algorithm [24], and the image of the meter area is intercepted. Then the intercepted meter area image is sent to the Deeplabv3+ [25] algorithm segmentation model for segmentation, and finally the meter reading is obtained after post-processing. The main contributions of this paper are as follows:By combining UAV and deep learning vision technology, the problems of the low efficiency and the high cost of traditional manual inspection or robot inspection are solved;The object detection algorithm YOLOv5s is introduced to improve the accuracy of detection of meter dial area and classification;Deeplabv3+ is used for image segmentation and this method improves the detection accuracy of the pointer and the scale line;Based on the image segmentation results, the concentric circle sampling method is proposed to flatten the dial to realize the reading of the dial image.

This study is outlined as follows: Section 2 shows the YOLOv5 algorithm structure, the Deeplabv3+ algorithm structure, and the post-processing method of the meter readings; Section 3 introduces various comparative experiments and experimental results from the same time and the experimental results are analyzed; Section 4 concludes the study; and Section 5 puts forward an outlook for the future in view of the shortcomings of the research.

## 2. Methods

### 2.1. Meter Reading Recognition Based on Object Detection and Image Segmentation

According to the characteristics of the UAV aerial photography substation meter image and the shortcomings of the traditional meter reading method, this paper adopted the object detection algorithm and semantic segmentation technology based on deep learning, which can accurately obtain the meter image and the meter corresponding to the image. The scale and pointer area realized the reading of the substation equipment. Object detection technology was used to detect the meter area in the image, which generally refers to the minimum outer-enclosing rectangular area of the closely surrounded meter target; the image segmentation technology is used to further segment the meter pointer and scale area target pixels in the meter image.

The idea of this paper was firstly to use the YOLOv5s object detection [26,27] technology to detect the area where the metered target is located in the UAV aerial image and eliminate the interference of the non-target area in the image; secondly, to accurately segment the intercepted meter image into the scale and pointer positions in the meter image by Deeplabv3+ image segmentation technology; and finally, to obtain the meter reading by post-processing. The processing flow of this paper is shown in Figure 1.

### 2.2. The YOLO Model

YOLOv5 [28] is a single-stage object detection algorithm. According to the depth and width of the network, YOLOv5 has four versions: YOLOv5s, YOLOv5m, YOLOv5l, and YOLOv5x. The depth of the network directly affects the detection accuracy and speed of the detector. The detection object in this paper was the aerial meter data and the detection object was small. On the premise of ensuring the detection accuracy, the detection model was installed on the edge device, so the YOLOv5s version would be used.

The YOLOv5s network consists of three parts and its network structure is shown in Figure 2. The backbone used mosaic data enhancement for splicing images through random scaling, random cropping, and random arrangement. The input image was imported into the Focus module for slicing operation and the sample slices with a scale of 640 × 640 × 3 were spliced into 320 × 320 × 12. At the same time, a Cross Stage Partial Networks (CSP) [29] structure and Spatial Pyramid Pooling (SPP) [30] were introduced to realize convolution and pooling down-sampling in order to extract the features. The next part was Neck, which was mainly the Feature Pyramid Networks (FPN) [31] and the Path Aggregation Network (PAN) [32]. The FPN layer transfered and fused the high-level strong semantic features through up-sampling from top to bottom. The PAN conveyed strong localization features from the bottom up and aggregated features from different backbone layers to different detection layers. The last part was the detection part, which took and output the feature maps of three scales, which are 17 × 17, 20 × 20, and 23 × 23, respectively. In the post-processing process, the model generated multiple anchor boxes based on the object features and used non-maximum suppression (NMS) [33]. If the confidence of the object being predicted as the object category was greater than the set threshold, it was retained, thus completing the object detection process.

YOLOv5s was used to detect the area where the meter was located within the image. In order to improve the accuracy of the final meter reading, this paper firstly detected the dial area of the image through the YOLOv5s network model. Using the labeling software, LabelImg, to make meter datasets, five different kinds of labels (bj, bjA, bjB, bjH, and bjL) in the meter images were defined. The type of meter represented by each label is shown in Figure 3. The algorithm itself automatically imported the training set into the YOLOv5s network model for training and generated the corresponding detection model weights.

### 2.3. Deeplabv3+ Split Tick Marks and Pointers

Deeplabv3+ is a well-polished state-of-the-art segmentation model, which has been widely used in many areas, such as the remote sensing area and medical image processing. This paper studied instrument segmentation detection. The proportion of the pointer and the scale was small, and the segmentation requirements were relatively fine. Deeplabv3+ has a better segmentation effect on fine objects, therefore, this paper chose Deeplabv3+.

Deeplabv3+ adopts a spatial pyramid pooling model and an encoding-decoding structure for semantic segmentation, it outputs feature map information in the upper backbone network, and it detects incoming features through a multi-rate and multi-effective field of view, which filters or pools operations to encode multi-scale features. The context information is enriched by encoding the semantic information and the decoder part gradually recovers the sharp target boundary information. In the meter detection images, because the pointer and the scale occupy a small proportion of the area where it is located, the segmentation of the image will be more difficult. The module structure of Deeplabv3+ is shown in Figure 4.

Compared with the Xception series used in the Deeplabv3+ paper as the backbone feature extraction network, a different backbone network, MobileNetv2 [34], was used in this paper. It is more suitable for deployment on edge devices than Xception and has more ideal and more efficient parameters and speed. The extracted feature maps went through the upgraded ASPP module. The feature map was first reduced in dimension through a 1 × 1 convolution kernel, then through three depthwise separable convolutions, and finally it was output through Adaptive Pooling. The compressed feature layer was passed into the decoder part through the backbone, and after being resized, it was concat with encode_data, and finally the output result was obtained through two convolution kernels and the Upsample.

In this paper, Deeplabv3+ was used to segment the tick marks and the pointer positions of the dial area. The training set of the segmentation network adopted the image of the dial area after being detected by the YOLOv5s detection network. Therefore, the cropped RGB image of the dial area was transmitted to the Deeplabv3+ network and it output the segmentation map as the same size as the input images. Figure 5 shows an example of the tick training set; the upper image is the original image, and the lower image is the label image.

### 2.4. Post-Processing Methods

#### 2.4.1. Erosion

The Deeplabv3+ segmentation removed most of the background and useless information, leaving only the scale and the pointer outline in the image; however, there were still many noises. In order to further eliminate interference, this paper performed erosion morphological processing on the segmented meter pointer and scale outline to remove the discrete point blocks. Assuming that the contour map point set is *A*, the convolution kernel is *B*, and *B* moves in order in *A*, the erosion image can be obtained. The segment image was obtained from the Formula (1):(1)$$A-B=\left\{x,y|{\left(B\right)}_{xy}\subseteq A\right\}$$

To remove the interference point blocks in the segmented image, the convolution kernel *B*, used in this paper, was designed as a 4 × 4 structure. The erosion operation eliminated the boundary points of the object, shrunk the boundary inward, and removed the objects that were smaller than the structural elements. The pointer and scale occupied fewer pixels in the segmentation map and were easily affected by noise. The effects of noise such as burrs and small bumps were removed by erosion. At the same time, two objects that are only connected by small blocks were disconnected to improve the reading accuracy of the meter.

#### 2.4.2. The Flattening Method and Meter Readings

After semantic segmentation and erosion processing, the circular dial area of the pointer image was flattened into a rectangular area by the concentric circle sampling method. The length and geometric center of the two sides of the divided rectangular image were used as the diameter and the center of the initial concentric circle, respectively, and the initial rotation angle and the width and height of the flattened rectangular area were specified. The initial rotation angle was used to generate the initial sampling point of each concentric circle. The width corresponded to the number of times of sampling for each concentric circle, and the height corresponded to the number of sampled concentric circles. Starting from the initial sampling point of the concentric circles, the pixel values were uniformly sampled on the circumference of the concentric circles in a clockwise direction. Taking the center of the concentric circles as the center, and shortening the radius by one pixel unit, a new concentric circle was generated. The steps for sampling were repeated several times, and finally the flattened rectangular area corresponding to the circular dial area was obtained.

The flattened rectangular area corresponded to the scale of the dial, and the midpoint coordinate of the line-segment was taken as the scale feature point for the scale, thereby forming a scale-center coordinate set that could represent the position of the scale. At the same time, the flattened image was scanned line by line from top to bottom, and the average value of the pointer pixel position was used as the coordinates of the pointer tip to indicate the pointer position.

Meter readings were accurately calculated by flattening the image, and the calculation formula is shown in (2):(2)$$R=\frac{\alpha }{\beta }\mu$$
where, *R* represents the meter reading, α represents the distance from the initial scale point of the flattened image to the pointer, β represents the distance from the initial scale of the flattened image to the end scale, and μ represents the total range of the meter.

### 2.5. Evaluation Indicators

Due to the complex environment of the substation, the aerial image of the UAV has a large receptive field, which will cause missed detection and false detection of the meter. Therefore, this paper used Precision and Recall to describe the meter detection model performance. The formulas of Precision and Recall are shown in (3) and (4):(3)$$\mathrm{Precision}=\;\frac{\mathrm{TP}}{\mathrm{TP}+\mathrm{FP}}$$
(4)$$\mathrm{Recall}=\frac{\mathrm{TP}}{\mathrm{TP}+\mathrm{FN}}$$

In the above formula, TP and FP represent the true and false positives, respectively, and FN represents the false negatives. In order to further evaluate the detection performance of the model, it was proposed to use the AP of a single category to represent the sum of the AP (average precision) values of each category, and to obtain the mAP (mean average precision) according to the AP value. The formulas for AP and mAP are shown in (5) and (6):(5)$$\mathrm{AP}=\int_{0}^{1}P\left(R\right)\mathrm{dR}$$
(6)$$\mathrm{mAP}=\frac{\sum_{q=1}^{Q}\mathrm{AP}\left(q\right)}{Q}$$

In Formula (6), *Q* is the number of categories.

The accuracy evaluation index of the image semantic segmentation model is expressed by mIoU (mean intersection over union), and the calculation formula is shown in (7):(7)$$\mathrm{mIoU}=\frac{1}{k}\sum_{i=1}^{k}\frac{p_{ii}}{\sum_{j=1}^{k}p_{ij}+\sum_{j=1}^{k}P_{ji}-P_{ii}}$$

Among them, K represents the number of label categories in the data set; $p_{ii}$ represents the number of the category label, i, in the data set, where the actual prediction is i*;* $p_{ij}$ represents the category label, j, where the actual predicted category is the number of i; and $p_{ji}$ represents the category label, i, where the actual predicted category is the number of j.

## 3. Experiment and Results

### 3.1. Experimental Conditions

#### 3.1.1. Data Acquisition and Transmission

Datasets were acquired by using the company’s self-produced drone nest, the developed platform scheduling software, and the DJI (Shenzhen DJI Sciences and Technologies Ltd., Shenzhen, China) Genie 4rtk (Real-Time Kinematic) UAV for data collection. The UAV itself integrates an HD video transmission system, a 360° rotating gimbal, and a 4K camera. The camera carried by the UAVs captured all the photos on an SD card. Transmission was carried out by a 4G/5G signal or, when it came back to the drone nest connected by LAN. All collected data was sent to the control center.

This paper collected the meter images in the Harbin substation and the filming hours were from 9:00 a.m to 6:00 p.m. The shooting environment included different weather patterns, lighting, and time periods, and the flight collection was specified according to the planned route. The flying height of the UAV was the same as the shooting point of the collecting meter, and the distance between the gimbal and the meter to be collected was about 1 to 1.5 m. Both the training set and the validation set are independent of each other. We collected a total of 1632 images in five different categories. There were 979 images in the training set and 653 images in the test set and the number of all the meter types is shown in Table 1.

#### 3.1.2. Experiment Platform

Server-side: Ubuntu 18.04, Intel^®^ Silver 4210 CPU@2.20 GHz, NVIDIA GeForce RTX A100(80 GB) GPU. The model framework is Pytorch 1.7.0, and the related software is CUDA 11.1, CUDNN 8.0.5 and Python 3.8.

Ubuntu18.04, Intel^®^ Xeon^®^ Gold 5120 CPU@2.20 GHZ, NVIDIA GeForce RTX 2080(8 G)*2 GPU. The model framework is PaddleX 1.3.3, and the related software is CUDA 10.2, CUDNN 8.1.0 and Python3.8.

### 3.2. Experimental Results 

#### 3.2.1. YOLOv5s Detection Results

After the training was completed, the YOLOv5s detection model needed to be tested before the semantic segmentation task, and the test results are shown in Figure 6. This shows that from different distances, different angles, and different weather conditions, the detection model could identify the position of the meter target in the image, and at the same time could correctly identify the meter type. This proves that the model used in this paper had a certain generalization ability and the training effect was ideal.

#### 3.2.2. Deeplabv3+ Image Segmentation Results

After the Deeplabv3+ network was trained, the network model was tested. Figure 7 shows the test results of the tick marks and the pointers of the model after network segmentation. The upper layer is the original scene image and the lower layer is the corresponding segmented image.

The test results show that the input image segmentation result was basically correct, and the position of the tick mark and the pointer could be accurately separated. The final image also needed to be corroded and flattened in order to correctly identify the dial reading.

#### 3.2.3. Flattening Results

In the rectangular area, the scales were evenly arranged from left to right, the lower end of the pointer was close to the center of the dial, and the upper end was close to the scale. Figure 8 shows, the result of flattening the image. The first scale coordinate of the scale center coordinate corresponded to the start scale position, and the last scale coordinate corresponded to the end scale position. The first distance between the pointer tip coordinate and the start scale position was calculated and the second distance between the end scale position and the start scale position was calculated. The ratio of the first distance to the second distance was multiplied by the total range of the type of meter to obtain the readings of several pointer meters in the substation.

### 3.3. Comparative Requirements

In order to choose the algorithm version that was more suitable for the research in this paper, a comparative requirement between the parameters, FLOPS and running speed of each version of YOLOv5 was carried out. The experimental results are shown in Table 2.

It can be seen from Table 2 that the YOLOv5s model had the least number of parameters and the fastest speed.

In order to verify the feasibility of using the detection and segmentation algorithm in this paper, this study compared and tested a variety of commonly used algorithms on the premise of the same dataset. The results of this comparison are shown in Table 3 and Table 4.

As can be seen in Table 3, this paper proposed the use of the YOLOv5s model with a faster detection speed for the detection of the meter dial of the aerial image. The Deeplabv3+ model with MobileNetV2 as the backbone network was used to segment the pointer and the scale of the meter image and the meter reading was realized through the meter post-processing technology. In this paper, the YOLOv5s model was used to detect a picture on the NVIDIA Geforce A100 GPU with an inference speed of 22 ms, which is significantly better than the YOLOv5m, YOLOv5L, YOLOv5X, YOLOv4 [35], and the YOLOv3 [36] models. The size of the model was only 14.1 MB; and mAP50 reached 99.584% in this dataset. The better detection accuracy and faster detection speed is able to meet the daily meter image inspection requirements.

In summary, compared with other commonly used detection algorithms, YOLOv5s has the smallest model and the fastest inference speed under the premise of ensuring model accuracy. It is more suitable for real-time detection on edge devices deployed on UAVs.

As can be seen from Table 4, for the five types of meters, the mIoU of the method used in this paper reached 78.92%, 76.15%, 79.12%, 81.17%, and 75.73%, respectively. It is obviously better than the Deeplabv1 model and the Deeplabv2 model, but not as good as than the original Deeplabv3+ model. However, the Deeplabv3+ model with MobileNetV2 as the backbone achieved a single image segmentation speed of 35.1 ms on NVIDIA Geforce 2080 GPU, which was significantly faster than the other three image segmentation models. The improved Deeplabv3+ model was only 11.1 MB and the parameters of the static model were only 2.8 MB. Furthermore, the model ran twice as fast, which greatly improved the model segmentation speed.

Table 4 shows that the Xception65 is the best backbone model, and this study did not need to achieve real-time detection. The method in this paper can be applied to substation meter reading whether using the Xception65 or the MobileNetv2 backbone network. Furthermore, the MobileNetv2 backbone network model detection speed was faster and we hope that the detection speed will be as fast as possible while still being of practical use.

Compared with the method of combining Faster R-CNN and U-Net in the literature [22], the method in this paper has the following advantages:Compared with the Faster R-CNN algorithm, the YOLOv5 algorithm was used in this paper, and the detection speed was significantly faster;The Deeplabv3+ image segmentation algorithm is mainly used in industrial applications, but the U-Net image segmentation method is mainly used for medical image segmentation, so it is better to use the Deeplabv3+ method for meter readings in industrial applications;The post-processing methods such as concentric circle sampling in this paper were more robust than the industrial applications in paper [22].

### 3.4. Meter Reading Interface Display

Figure 9 shows the meter reading results of this method, which are 1.2544, 0.4285, 0.3073, 0.0000, and 0.3977, respectively. It can be seen from the above that the method in this paper could accurately read the value of the meter image, providing an effective and accurate reading method for the UAVs aerial photography of the meter image, which, in this case, used a pointer to indicate the value.

### 3.5. Comparing Readings

Figure 10 shows the images of the five meters and Table 5 shows the manually measured values compared to the recognized values. The error shows the absolute values of the manually measured values minus the recognized values, which are 0.0212, 0.0002, 0.0184, 0.0330, and 0.0017, respectively. The effectiveness of this method in automatic identification and reading of substation meter images in UAV aerial photography is, therefore, proven.

The method proposed in this paper still has many aspects that need to be improved. We mainly realize that the reading detection of various pointer meters, and the provision of technical support for the inspection of substation meters, is needed. However, there are still many different types of meters that have not been studied, and when the area of the meter is not complete due to the external conditions, it will cause errors or lead to large reading errors.

## 4. Conclusions

This paper has designed a method that combined YOLOv5s and Deeplabv3+ and has implemented a substation meter detection and reading method through a series of post-processing methods. The test results have proven that the method proposed in this paper could accurately read various meter types at different angles and under different conditions. The main contributions of this paper are as followsThe use of UAVs to fly through designated routes at different times and different weather conditions and the collection of 1632 images, including five different types of meters for object detection model training;The improvement of: the backbone network of the Deeplabv3+ semantic segmentation network; and the inference speed of the segmentation algorithm for a single image, which was twice the speed of the original model and a reduction in the size of the model weight;The use of the erosion and concentric circle sampling method to flatten images to realize meter panel reading. The result has been to achieve an accurate reading of the meter readings while quickly detecting the meter area. In this paper, the inspection of substation instruments was combined with deep learning visual algorithms and mobile flying equipment. It is hoped that the work in this paper can provide some help for intelligent substation inspection.

## 5. Future Work

The main future work would be to continue to improve detection accuracy and speed, especially for different kinds of meters and more complex background conditions. At present, the intelligent inspection technology of meter readings has become an important development direction of the intelligent inspection of substations. In future research, we will increase the detection and segmentation of other components in the substation environment, and at the same time combine other algorithms such as object tracking and key point detection to achieve state estimation and prediction, which will further improve the intelligence level of substation inspection from all aspects.

## Figures and Tables

**Figure 1 sensors-22-07090-f001:**
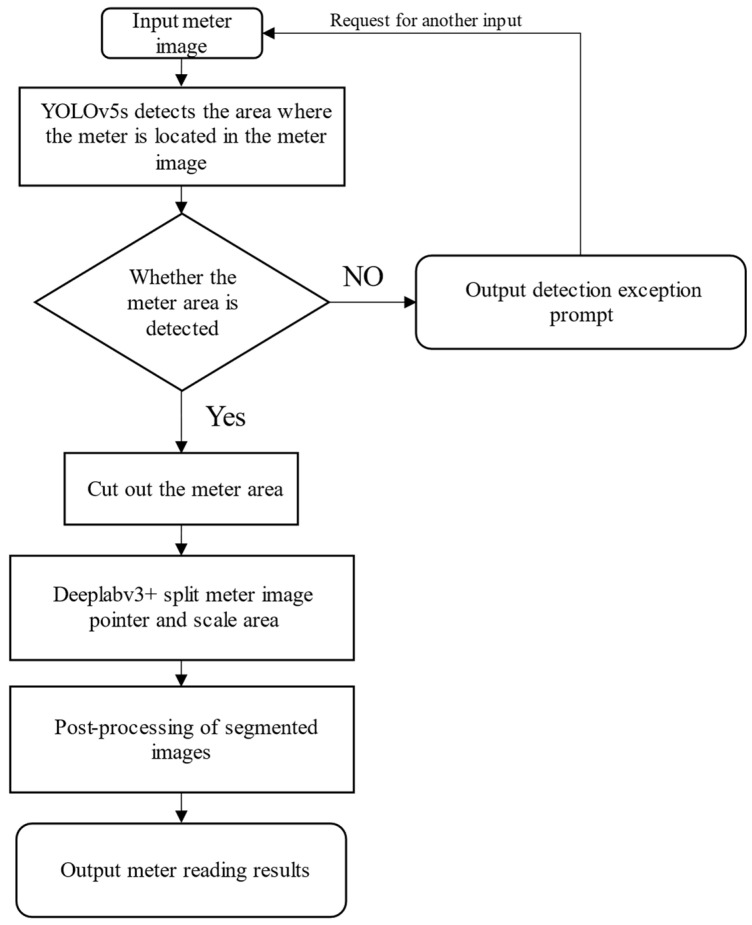
The process of identifying meter readings based on YOLOv5s and Deeplabv3+.

**Figure 2 sensors-22-07090-f002:**
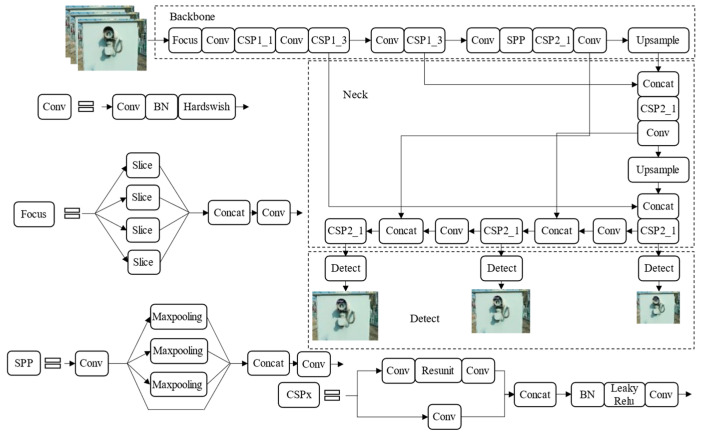
The YOLOv5s network structure diagram.

**Figure 3 sensors-22-07090-f003:**

Five kinds of meters. (**a**) bj, which refers to oil level, (**b**) bjA, which refers to Sulfur Hexafluoride Density Relay, (**c**) bjB, which refers to Discharge Counter With Current Meter For Arrester, (**d**) bjH, which refers to Discharge Counter and (**e**) bjL, which also is Sulfur Hexafluoride Density Relay.

**Figure 4 sensors-22-07090-f004:**
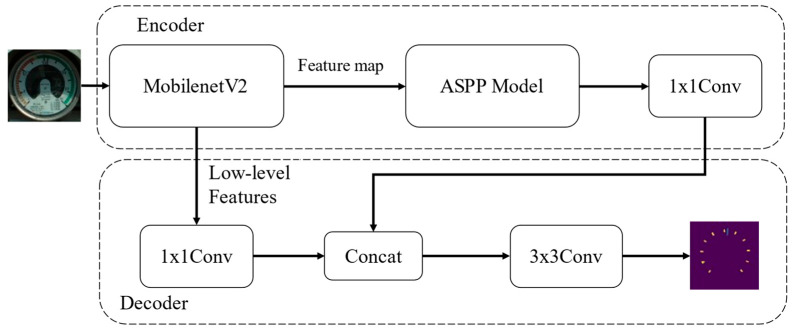
The Modules of Deeplabv3+.

**Figure 5 sensors-22-07090-f005:**
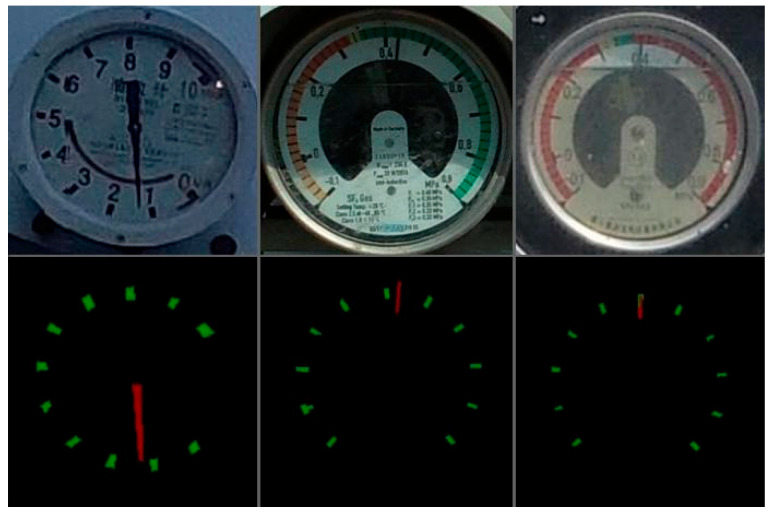
An example of segment datasets.

**Figure 6 sensors-22-07090-f006:**
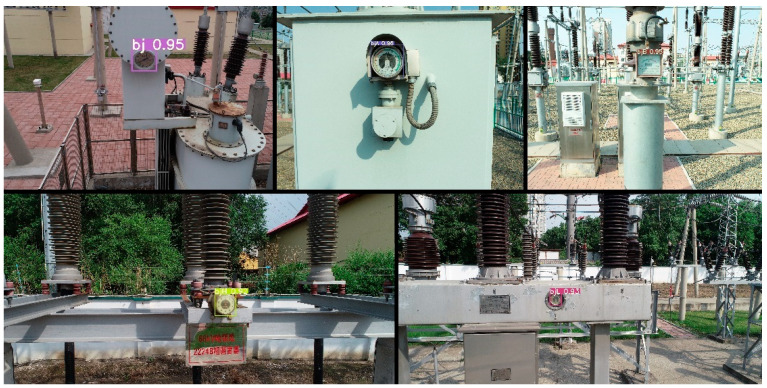
Dial detection results using YOLOv5s.

**Figure 7 sensors-22-07090-f007:**
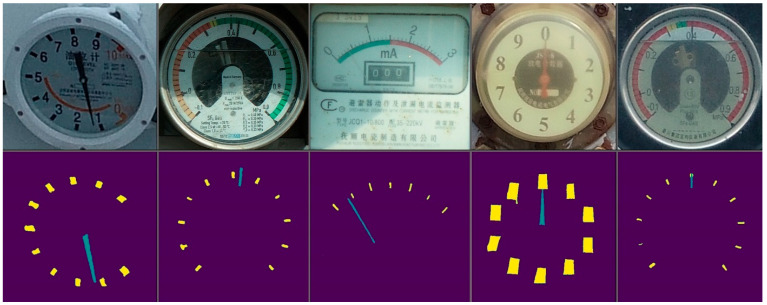
Image segmentation results of Deeplabv3+.

**Figure 8 sensors-22-07090-f008:**
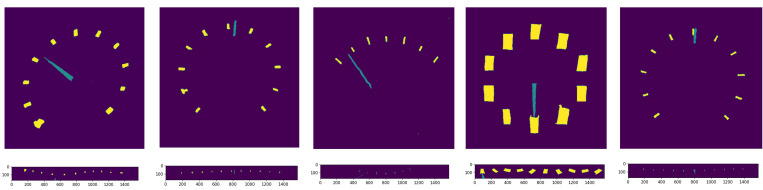
The result of flattening the image.

**Figure 9 sensors-22-07090-f009:**
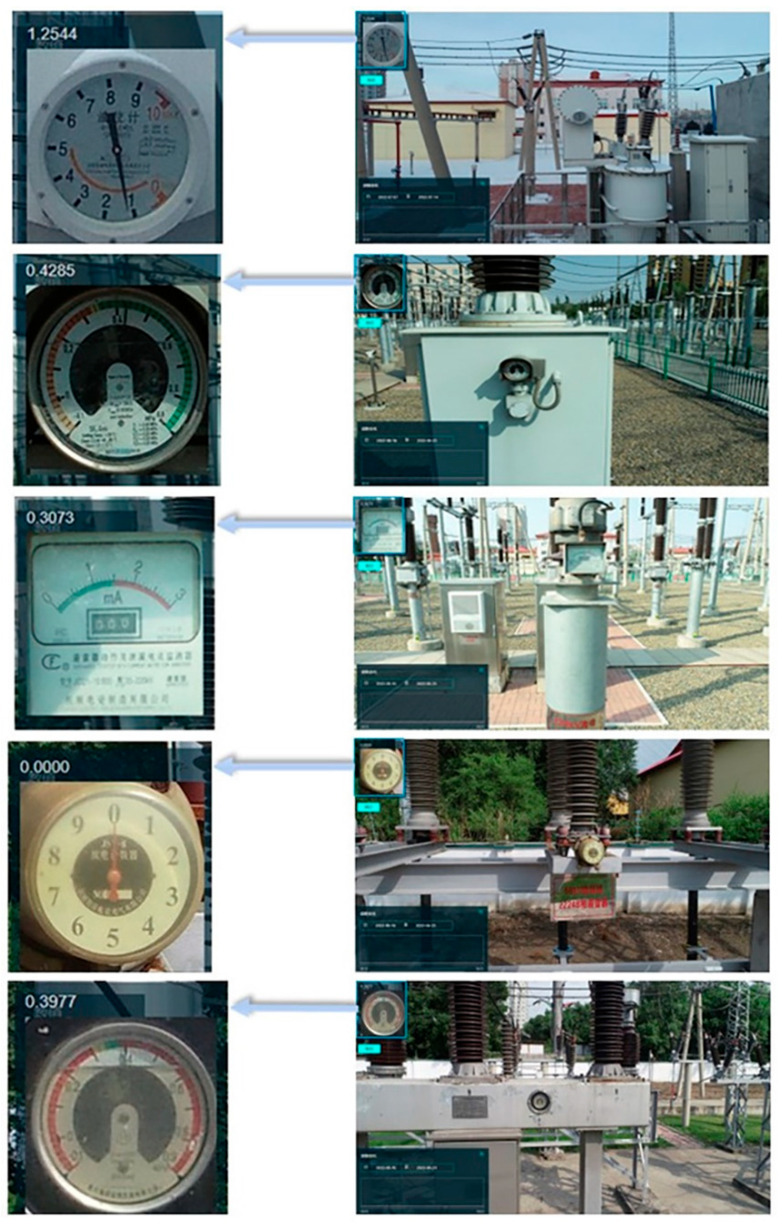
The results of the meter reading.

**Figure 10 sensors-22-07090-f010:**
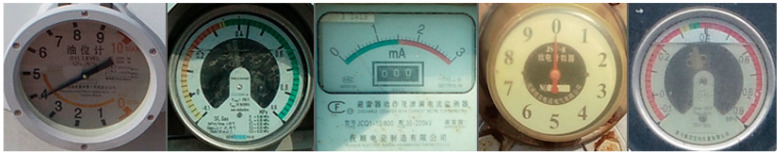
Meter images.

**Table 1 sensors-22-07090-t001:** Dataset Statistics.

	bj	bjA	bjB	bjH	bjL
Training set	75	329	301	126	146
Test set	58	213	179	106	97

**Table 2 sensors-22-07090-t002:** A comparison of yolov5 detection model results.

Model	Speed/ms	FPS	Params	FLOPS
YOLOv5s	22.2	45.0	7.5M	13.2B
YOLOv5m	27.0	37.0	21.8M	39.4B
YOLOv5l	29.2	34.2	47.8M	88.1B
YOLOv5x	30.8	32.5	89.0M	166.4B

**Table 3 sensors-22-07090-t003:** The comparison of meter detection model results.

	bj/%	bjA/%	bjB/%	bjH/%	bjL/%	mAP50/%	Speed/ms	Model Size/MB
YOLOv3	99.536	99.623	99.592	99.575	99.567	99.579	27.4	123.4
YOLOv4	99.538	99.610	99.571	99.561	99.566	99.569	34.0	256.3
YOLOv5X	99.540	99.626	99.593	99.576	99.571	99.581	30.8	177.5
YOLOv5L	99.539	99.613	99.584	99.575	99.570	99.576	29.2	90.8
YOLOv5m	99.542	99.630	99.600	99.579	99.573	99.585	27.0	41.3
YOLOv5s	99.542	99.628	99.599	99.579	99.571	99.584	22.2	14.1

**Table 4 sensors-22-07090-t004:** A comparison of segmentation results.

	Backbone	bj_mIoU_/%	bjA_mIoU_/%	bjB_mIoU_/%	bjH_mIoU_/%	bjL_mIoU_/%	Speed/ms	Model Size/MB
Deeplabv1	VGG16	61.69	52.35	44.54	67.01	37.33	15.8	82.0
Deeplabv2	Resnet101	57.74	47.12	33.55	77.09	45.06	56.0	176.9
Deeplabv3+	Xception65	85.62	76.95	82.14	82.93	80.03	66.8	165.1
Deeplabv3+	MobileNetV2	78.92	76.15	79.12	81.17	75.73	35.1	11.1

**Table 5 sensors-22-07090-t005:** Automated recognized results compared to manually read values.

	Manually Measured Values	Recognized Values	Error
bj	3.8500	3.8288	0.0212
bjA	0.4385	0.4383	0.0002
bjB	0.3900	0.3716	0.0184
bjH	0.0000	0.0330	0.0330
bjL	0.4055	0.4072	0.0017

## Data Availability

Not applicable.

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
