# Peer review of "Automatic Meter Reading from UAV Inspection Photos in the Substation by Combining YOLOv5s and DeeplabV3+"

_sensors, 2022, doi:10.3390/s22187090_

Round 1

Reviewer 1 Report

This study proposes a method based on the combination of YOLOv5s object detection and Deeplabv3+ image segmentation for obtaining the meter readings from segmented images for meters with a pointer dials.
The aerial images were collected using the UAV as compared to previous methods where static cameras were installed at the substations. Overall the problem presented is interesting and the proposed approach has merits.

However, there are quite a few points which need to be addressed:

It is not clear in lines 24-25 what "the image segmentation model mIoU reaches 78.92%, 76.15%, 79.12%, 81.17% and 75.73% respectively" is referring to. For what labels are these values computed?

The process of capturing the aerial images and transmitting them to the control center is not clearly described (lines 79 - 84). How far is the control center from the places of image collection? what is the effect of this distance on transmission and detection? Do you need real-time detection? If not, then what's the importance of latency? How is the camera set on UAV? What is the setting of meters at the substations? Did all meters have the same range? How about working with meters with different ranges? The overall automation of computing the final readings is vaguely described in methodology and results sections.

The results from the YOLOv5 are quite accurate because of the robust algorithm. However, the most challenging part for computing the exact readings is the segmentation where the performance is still not up to the mark.
As per Fig. 1, what is the course of action when the meter area is not detected?

Line 144 - What are these five different kinds of labels (bj, bjA, bjB, bjH, bjL) in the meter images? No description has been provided till this point.

There are number of incomplete sentences in the manuscript such as:

Line 45-46 - incomplete sentence
Line 80 - incomplete sentence
Lines 145-146 - incomplete sentence

Reviewer 2 Report

Dear Editor and Dear Authors,

The topic of this paper is very interesting and thank you for your trust to send me the manuscript to review it.

The manuscript proposes a method based on the combination of YOLOv5s object detection and Deeplabv3+ image segmentation, and obtains meter readings through post-processing of segmented images. Authors state that the results show, that the method in this paper significantly improves the accuracy and practicability of substation meter reading detection in complex situations.

The comments and suggestions can be find below:

1.      The Abstract is well written.

2.      The Introduction section should be separated from the related work overview. The related work should be in separate section with valuable references, after the Introduction section. Also, the brief overview of the paper by sections should be given in the end of the Introduction section for better clarification of the concept of the paper itself.

3.      All the abbreviations should be explained in the paper.

4.      The meanings and differences of YOLOv5s, YOLOv5m, YOLOv5l, YOLOv5x should be explained with more details, so that interested readers who are new to this topic would understand more easily what it is about. Further, the authors should explain why did they choose YOLOv5 for the detection, and what are the main advantages and drawbacks of this object detector architecture.

5.      Deeplabv3+ is a great choice for the segmentation, but authors should explain why did the choose it, what are the main properties of this segmentation network.

6.      The results are well presented and the examples are illustrative. Maybe some comparisons could be given after the experiments part.

7.      The discussion should be placed in the end of the experiments section, and the future works part should be brief after the Conclusion section.

8.      I think the conclusions part is too long, because it should briefly reflect the results of the research itself. This section should be concise, not to long in my opinion.

The paper is very interesting from the practical point of view. It is well written, however revision is needed. After the appropriate revision and correction the paper can be published.

Recommendation: Accept after the appropriate revision.

Reviewer 3 Report

It is really good work, it combines Yolo and DeepLab, my suggestions are:

A)You could upload your code to Github;

B) You could upload the dataset to some repository, e.g. Zenodo;

C) add more details of the collected dataset and how you split between training and test set, it is not possible to understand if the dataset is split in a fair way (e.g. images of the same meter should be no split between training and test set);

E) ‘Figure 6’  check the fourth column, the result of the segmentation is clearly wrong (see the red needle);

F) Is the performance reported in table 2 before or after the post-processing? If 'after' then report also the performance 'before', to understand how much it is useful;

G)   In table 3 is clear that the best backbone is Xception65, why do you prefer mobile net? Your application is not a real-time problem, moreover, the model size is not a problem with the current GPU. Do I miss something? Why do you test different backbones for the different DeepLab versions? Please add some comments. 

H) in the introduction you should better describe the current state of the art in detection and segmentation, for example, I suggest adding:

https://arxiv.org/abs/2106.13797

https://arxiv.org/abs/2207.02696

https://www.mdpi.com/2624-6120/3/2/22

Round 2

Reviewer 1 Report

I am fine with the corrections made in the manuscript.

Line 25: Is it "the image segmentation model" or "models"?

Reviewer 3 Report

Revision well done